# A Novel Spectral Index to Identify Cacti in the Sonoran Desert at Multiple Scales Using Multi-Sensor Hyperspectral Data Acquisitions

Kyle Hartfield [1,*], Jeffrey K. Gillan [1], Cynthia L. Norton [1], Charles Conley [1] and Willem J. D. van Leeuwen [2]

[1]  University of Arizona School of Natural Resources and the Environment, Arizona Remote Sensing Center, The University of Arizona, 1064 E. Lowell Street, Tucson, AZ 85721, USA; jgillan@email.arizona.edu (J.K.G.); cnorton1@email.arizona.edu (C.L.N.); charlesconley@email.arizona.edu (C.C.)
[2]  University of Arizona School of Natural Resources and the Environment and School of Geography, Development & Environment, Arizona Remote Sensing Center, The University of Arizona, 1064 E. Lowell Street, Tucson, AZ 85721, USA; leeuw@email.arizona.edu
*  Correspondence: kah7@email.arizona.edu

**Abstract:** Accurate identification of cacti, whether seen as an indicator of ecosystem health or an invasive menace, is important. Technological improvements in hyperspectral remote sensing systems with high spatial resolutions make it possible to now monitor cacti around the world. Cacti produce a unique spectral signature because of their morphological and anatomical characteristics. We demonstrate in this paper that we can leverage a reflectance dip around 972 nm, due to cacti's morphological structure, to distinguish cacti vegetation from non-cacti vegetation in a desert landscape. We also show the ability to calculate two normalized vegetation indices that highlight cacti. Furthermore, we explore the impacts of spatial resolution by presenting spectral signatures from cacti samples taken with a handheld field spectroradiometer, drone-based hyperspectral sensor, and aerial hyperspectral sensor. These cacti indices will help measure baseline levels of cacti around the world and examine changes due to climate, disturbance, and management influences.

**Keywords:** hyperspectral; cacti; drone

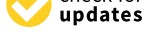



## 1. Introduction

The Cactaceae (cacti) family is one of the most threatened plant families on the planet, while being some of the most important flora in arid regions of the North American continent [1,2]. A variety of mammals, birds, and insects rely on cacti for shelter and as a source of nutrients and hydration during the hot summer season [3–10]. People also use various cacti for ornamental horticulture, food, and medicinal purposes [2]. Facing pressure from land conversion for agriculture, horticulture collection, and urban development, nearly a third of cacti worldwide fall into the threatened category [2]. Cacti also face threats from climate change as arid regions become more arid [11,12]. Although cacti are adapted to survive in areas with limited precipitation, extended periods of drought and increases in summer temperatures harm the establishment of seedlings that require moist soils to flourish [11,13].

In other regions of the world, cacti are an invasive species that threatens native plants. In Kenya, degraded rangelands overrun with prickly pear impact forage for both wildlife and livestock [14,15]. Ranchers in the Edwards Plateau region of Texas work to determine the best strategy to control prickly pear encroachment on fire-disturbed rangelands [16]. Research shows clusters of invasive cacti in South Africa, Australia, and Spain due to ideal climates for cacti. Climate change trends also indicate that parts of China, eastern Asia, and central Africa are suitable for future cacti invasions [17].

Despite the recognized need to monitor cacti occurrence and density, information on cacti population trends are relatively unknown over large areas [2]. The limited geographic

extent of field surveys (e.g., transects and plots) makes remote sensing a useful approach for mapping cacti. Other studies have demonstrated cacti mapping with the use of convolution neural networks from drone imagery [18] and random forest supervised classification with Sentinel 2 satellite imagery [15]. These methods may be challenging in the southwestern United States, where individual cacti plants are often much smaller than satellite imagery spatial resolution. Additionally, multispectral satellite imagery will have a difficult time separating cacti from other spectrally similar plant species.

A potential key to distinguishing cacti from non-cacti plants lies in their morphological differences and how they perform photosynthesis. Cacti contain cells designed to store water long term and exhibit a form of photosynthesis allowing their stomata to open at night to limit water loss through transpiration. Most cacti do not have leaves and perform photosynthesis with the tissue layer of their stems [19–21]. These structural differences influence the fate of solar radiation in nuanced ways, best observed with hyperspectral imaging sensors [22–25]. Previous research has shown that saguaro cacti (*Carnegiea gigantea*) exhibit a strong dip in near-infrared reflectance values around the 970 nm due to water absorption [24]. Other research showed the utility of using measurements at 970 nm to estimate plant water concentration [26].

This paper had two main objectives:

1. Using hyperspectral field spectroradiometer measurements to examine the spectral signatures of cacti and non-cacti desert adapted plants to find distinguishing characteristics that would allow for the development of a spectral 'cacti index'.
2. Examine the efficacy of cacti signatures and the index to identify cacti from a drone-mounted hyperspectral sensor (3 cm resolution) and an airplane-mounted hyperspectral sensor (1 m resolution).

Accurate identification of cacti will allow us to monitor extent baselines and changes due to climate variability, climate change, disturbance events, and other human-related modifications of the environment.

## 2. Materials and Methods

### 2.1. Study Area

Our study used data collected in the Santa Rita Experimental Range (SRER) located about 30 km south of Tucson, AZ (Figure 1). SRER, founded in 1903, is the longest continuously active rangeland research facility in the United States [27]. The National Science Foundation (NSF) National Ecological Observation Network (NEON) has designated SRER as a terrestrial core site [28].

SRER is located in a semi-arid ecosystem with a bimodal precipitation distribution. The area receives 28 to 50 cm of rain annually with the majority of events occurring in the winter and summer. The average annual temperature is 20 °C. Cacti are common in the area due to a lack of extended freezing temperatures during the winter months [27].

Elevation levels range from 900 m in the northwest to 1400 m in the southeast. The vegetation composition consists of creosote (*Larrea tridentata*), prickly pear (*Opuntia engelmannii*), cholla (*Cylindropuntia fulgida*), barrel cactus (*Ferocactus wislizeni*), mesquite (*Prosopis velutina*), palo verde (*genus Parkinsonia*), whitethorn acacia (*Acacia constricta*), yucca (*Yucca rostrata*), lotebush (*Ziziphus obtusifolia*), and various grasses [27].

The airplane-mounted hyperspectral AVIRIS (Airborne Visible-Infrared Imaging Spectrometer) exploration occurred in the northwest part of SRER, an ideal environment for cacti observed through a high occurrence of prickly pear and cholla cacti (Figure 1). The drone-based hyperspectral analysis occurred closer to the center of the SRER at an elevation of 1130 m. This area had plentiful prickly pear and barrel cacti. Due to the higher elevation and monsoon rains (collected in August 2021), the area had a substantial herbaceous cover of grasses and forbs.

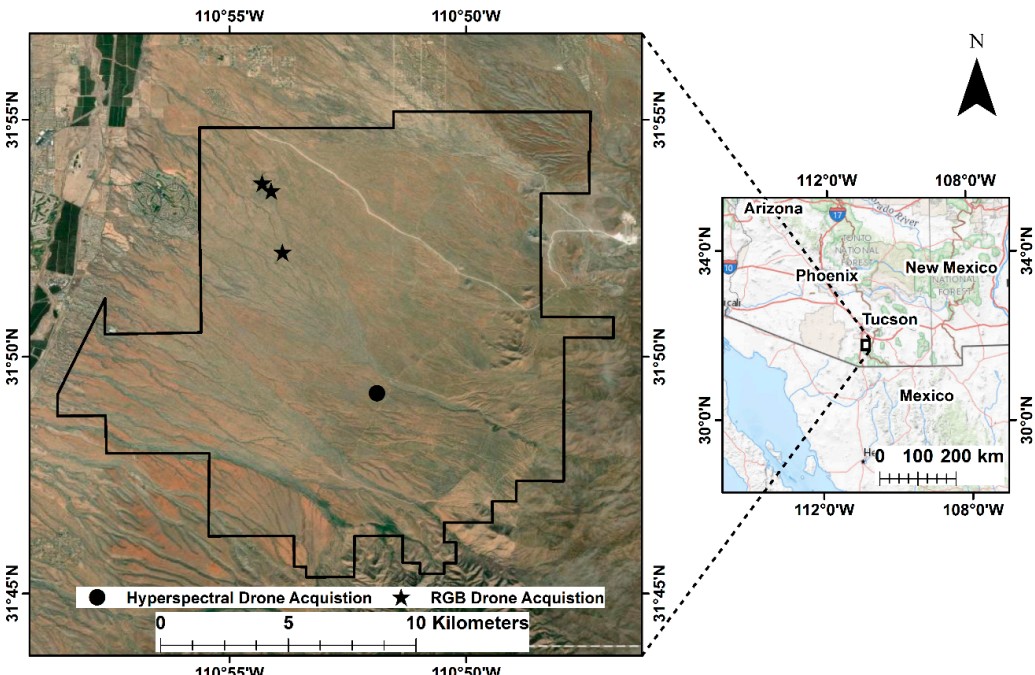

**Figure 1.** We conducted this study at the Santa Rita Experimental Range (SRER), located about 30 km south of Tucson, Arizona. We identified cacti samples for the NEON AVIRIS imagery analysis using photos acquired with a DJI Mavic Pro multirotor in the northwest portion of the study area (black stars), an area with a high density of cacti. We collected drone-mounted hyperspectral imagery in the center of the SRER, where prickly pear and barrel cacti are plentiful (black circle).

*2.2. Data and Methods*

2.2.1. Field Spectroradiometer

We acquired outdoor clear-sky hyperspectral nadir measurements with a full-range (350–2500 nm) spectroradiometer, FieldSpec®3 by Analytical Spectral Devices Inc. (ASD) Boulder, CO, USA, with a 15-degree field of view during the summer of 2017. We used a calibrated Spectralon panel to compute component/surface reflectance data [24]. We took measurements from 5 chollas, 11 prickly pear cacti, 9 barrel cacti, 32 mesquite, 9 creosote, 26 patches of bare soil, and 28 plots of grass. We averaged the measurements for each type of sample.

From the field spectroradiometer data, we identified the same reflectance dip at 972 nm first reported by van Leeuwen. This reflectance dip is due to water absorption in succulent cacti and can serve as a distinguishing characteristic from non-cacti vegetation [24].

We measured the magnitude of the dip with a normalized difference approach between the reflectance values at the bottom of the dip and reflectance values immediately outside of the dip. This is similar to the concept of the normalized difference vegetation index (NDVI) [29].

A priori, it was unknown which specific bands outside of the dip would provide a robust index capable of distinguishing cacti from non-cacti vegetation. We tested two different spectral indices.

Cacti Index 1 (CI1) uses reflectance at 862 nm, which occurs immediately before the dip at 972 nm:

$$\text{Cacti Index 1} = \frac{862\,\text{nm} - 972\,\text{nm}}{862\,\text{nm} + 972\,\text{nm}}$$

The second equation, Cacti Index 2 (CI2), uses reflectance at 1072 nm, which occurs immediately after the dip at 972 nm:

$$\text{Cacti Index 2} = \frac{1072\,\text{nm} - 972\,\text{nm}}{1072\,\text{nm} + 972\,\text{nm}}$$

2.2.2. Taking the Cacti Index Airborne

Using the cacti index with aerial and drone-acquired imagery would greatly improve its utility for mapping and monitoring cacti populations across large tracts of land. We also wanted to examine the impact of spatial resolution on the uniqueness and clarity of the cacti signature. We tested the efficacy of the cacti index using both drone- (3 cm spatial resolution) and airplane-mounted (1 m spatial resolution) hyperspectral sensors.

To investigate the utility of the cacti index with drone scale imagery, we used a Nano Hyperspectral (https://www.headwallphotonics.com/products/vnir-400-1000nm (accessed on 23 May 2022)) Visual & Near Infrared (VNIR) sensor by Headwall Photonics gimbal mounted on a DJI Matrice 600 Pro 6 rotor copter, which is a push broom slit sensor with 640 linear array detectors. The data contain 270 bands ranging from 400 nm (blue) to 1000 nm (near infrared) at ~2.2 nm slices. In August 2021, we collected imagery over a 2 ha plot in SRER known to have a mix of cacti and non-cacti vegetation. We flew the drone ~65 m above ground level, yielding a spatial resolution of ~3 cm. Using Headwall software, we converted the raw imagery digital numbers to radiance and then to reflectance using a tarp with known reflectance values. Individual frames were then orthorectified and mosaicked into a stacked imagery product. On the stacked orthomosaic, we identified and extracted spectral signatures from 11 prickly pear, 7 mesquite, 10 barrel cacti, 10 patches of bare ground, and 10 samples of grasses. Due to known sensor noise in the near-infrared region of the spectrum, we employed a three-band moving average to smooth the spectral signatures. We calculated CI1 using bands 212 (864 nm) and 260 (970 nm) and extracted these values from the same vegetation samples. The sensor is not sensitive to radiation beyond 1000 nm, so we were unable to calculate CI2.

To investigate the utility of the Cacti Indices at an airplane imagery scale, we used NEON Airborne Observation Platform hyperspectral data collected across the SRER between August 24 and 29 of 2018. The sensor was a next-gen version of the Airborne Visible/Infrared Imaging Spectrometer (AVIRIS-NG) (https://avirisng.jpl.nasa.gov/aviris-ng.html (accessed on 23 May 2022)). The data contain 426 bands ranging from 380 nm (blue) to 2500 nm (near infrared) at ~5.5 nm slices. The aircraft flew ~1000 m above ground level with a nominal spatial resolution of 1 m. NEON used ATCOR4r [30] to atmospherically correct the data and serve it as unitless surface reflectance values scaled by 10,000.

To facilitate the identification of individual cacti and non-cacti vegetation samples, we collected 35 ha of high-resolution (1.6 cm) RGB drone imagery using a DJI Mavic Pro multirotor. We found this method to be more efficient than locating samples on foot. We co-registered orthomosaics created from the drone imagery with the NEON AVIRIS imagery using ArcGIS Pro. We identified and extracted the spectral signature from many samples of cholla (N = 418), prickly pear (N = 325), mesquite (N = 105), palo verde (N = 100), creosote (N = 100), and bare ground (N = 300). We calculated CI1 and CI2 using AVIRIS bands 97 (862 nm), 119 (972 nm), and 139 (1072 nm).

## 3. Results

### 3.1. Spectral Signatures from Ground-, Drone-, and Airplane-Based Sensors

Spectral signatures for all three sensors showed similar characteristics but nuanced differences. The prickly pear, cholla, and barrel cactus spectral signatures, as measured from the ASD spectroradiometer, all exhibit a dip in reflectance centered at 972 nm and a peak around 1072 nm (Figure 2). The non-cacti classes do not have this dip. The prickly pear and cholla signatures show a similar dip at 972 nm and a peak at 1072 nm when extracted from the NEON AVIRIS data. However, the absolute reflectance values are lower and the dip is shallower. From the drone-mounted Nano Hyperspectral sensor, the same water absorption dip at 972 nm is present in the barrel and prickly pear cacti signatures, but the absolute reflectance values are also lower than the ASD.

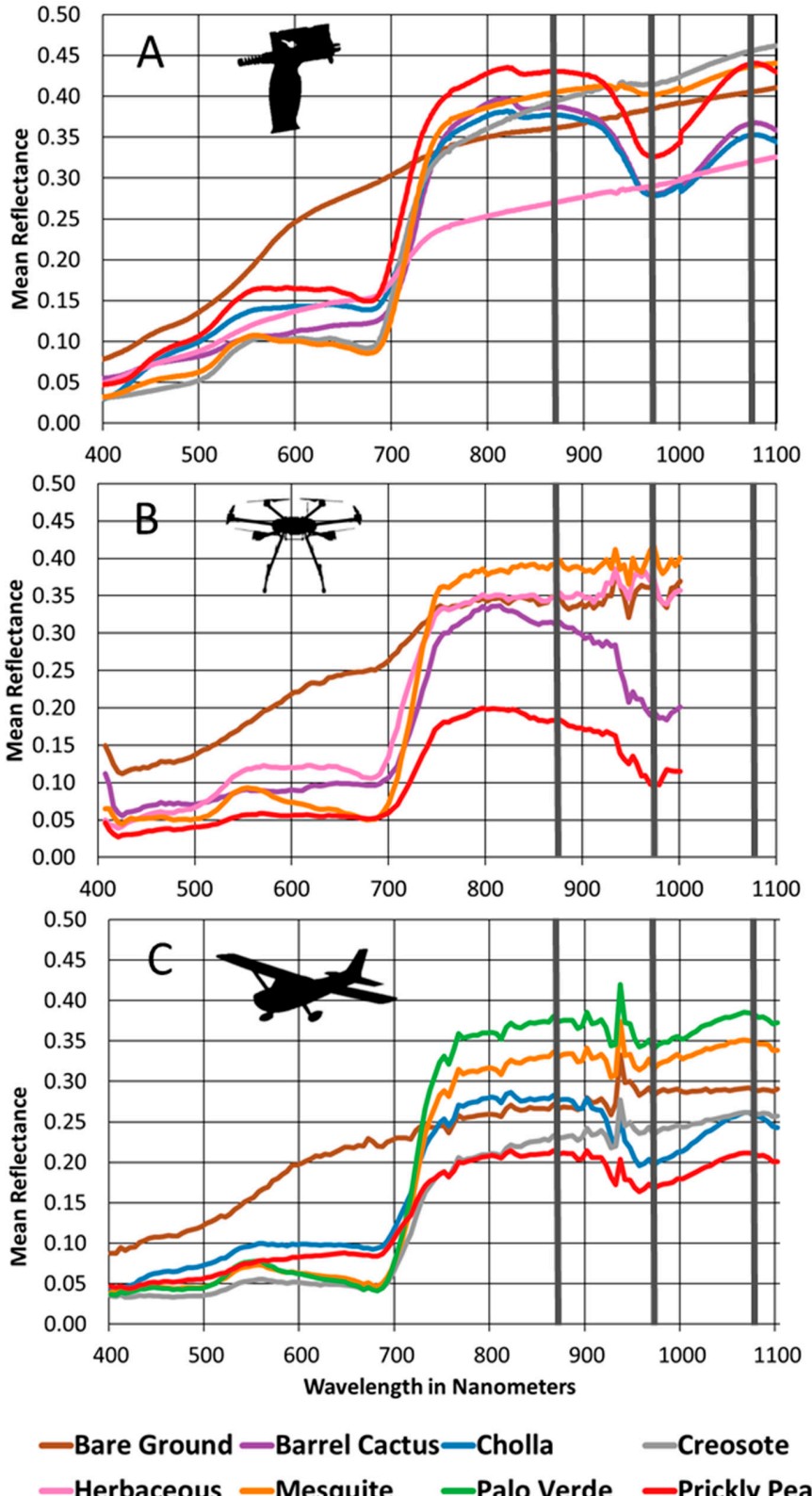

**Figure 2.** Average spectral signatures of mesquite (orange line), creosote (grey line), bare ground (brown line), herbaceous (pink line), barrel cactus (purple line), cholla (blue line), and prickly pear (red line), as measured from by the ASD spectroradiometer (**A**), drone-mounted Nano Hyperspectral sensor (**B**), and airplane-mounted NEON AVIRIS (**C**). The three dark grey lines represent the three portions of the electromagnetic spectrum used to calculate the two cacti indices.

### 3.2. Range of Cacti Indices by Plant Type

Using the field spectroradiometer, we sampled bare ground (N = 26), barrel cactus (N = 9), cholla (N = 5), creosote (N = 9), herbaceous (N = 28), mesquite (N = 32), and prickly pear (N = 11). CI1 values calculated from the field spectroradiometer show a separation of the cacti from non-cacti vegetation and bare ground (Figure 3). The majority of CI1 values for the cholla samples range from 0.173 to 0.186, with a mean value of 0.133. The majority of CI1 values for prickly pear samples range from 0.095 to 0.177, with a mean value of 0.103. The majority of CI1 values for the barrel cactus samples range from 0.101 to 0.257 with a mean value of 0.125.

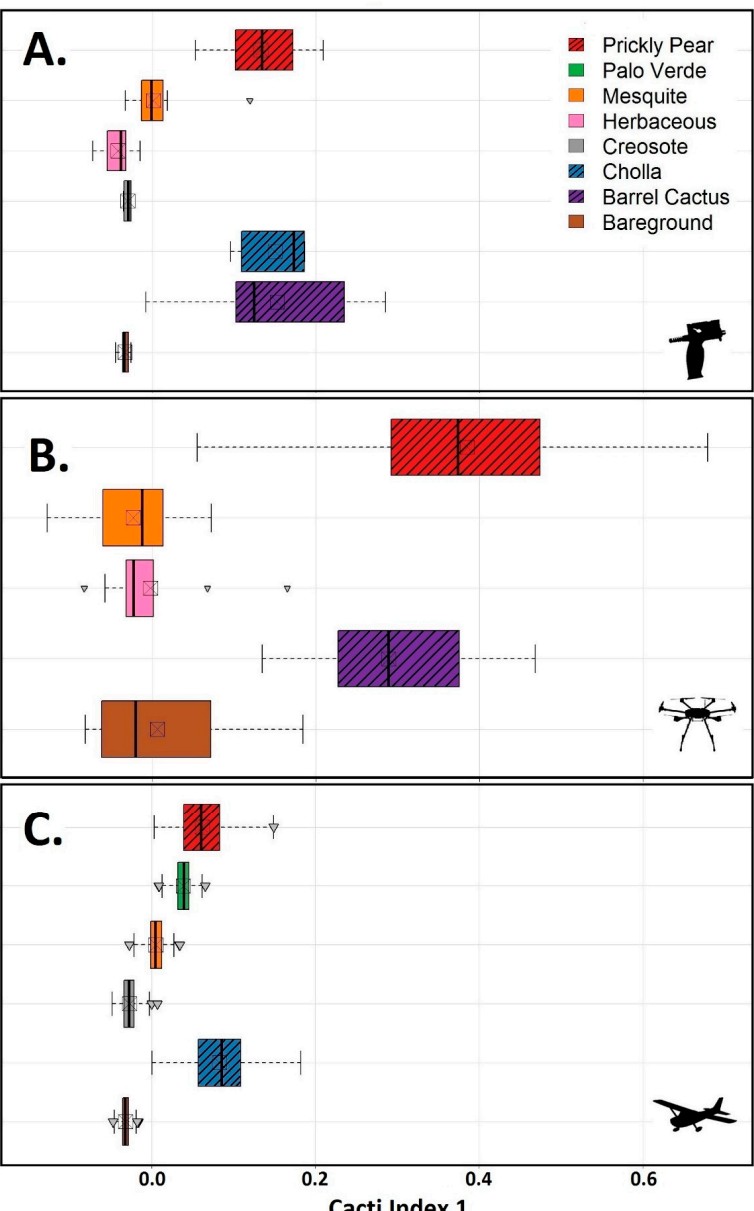

**Figure 3.** This graph shows the range of Cacti Index 1 (862 and 972 nm) values for cacti, non-cacti vegetation, and bare ground computed from the Field Spectrometer (**A**), Nano Hyperspectral (**B**), and NEON AVIRIS (**C**) collections. The red hatched boxes represent prickly pear, the blue hatched boxes represent cholla, and the purple hatched boxes represent barrel cactus. The square in the middle of each box is the mean for that series of samples. The whiskers represent the minimum and maximum values within the interquartile range. Any triangles outside the whiskers are outlier values.

We identified bare ground (N = 10), barrel cactus (N = 10), herbaceous (N = 10), mesquite (N = 7), and prickly pear (N = 11) using the drone-mounted Nano hyperspectral sensor. CI1 values calculated using the Nano hyperspectral data for the selected samples show a separation of the barrel cactus and prickly pear (cacti) from non-cacti vegetation and bare ground. The majority of values for the barrel cactus samples range from 0.205 to 0.390, while the majority of values for the prickly pear samples range from 0.264 to 0.482. The mean value for the barrel cactus samples is 0.289 and the mean value for the prickly pear samples is 0.385 (Figure 3).

The greater coverage of the NEON AVIRIS data made it possible for us to recognize bare ground (N = 300), creosote (N = 418), herbaceous (N = 100), mesquite (N = 105), palo verde (N = 100), and prickly pear (N = 325). CI1 values calculated using the NEON AVIRIS data for the selected pixels show a separation of the cholla and prickly pear (cacti) from non-cacti vegetation and bare ground. The majority of values for the cholla samples range from 0.056 to 0.109, while the majority of values for the prickly pear samples range from 0.039 to 0.083 on the CI1. The mean value for the cholla samples is 0.083 and the mean value for the prickly pear samples is 0.062. The only other series of samples that overlaps the two cacti boxes are those for palo verde with a range from 0.032 to 0.045 and a mean of 0.040 (Figure 3).

We used the same bare ground (N = 26), barrel cactus (N = 9), cholla (N = 5), creosote (N = 9), herbaceous (N = 28), mesquite (N = 32), and prickly pear (N = 11) samples pulled from the field spectroradiometer to investigate CI2. The CI2 values show a separation of cacti from the other land cover types (Figure 4). The mean value for the cholla samples is 0.083 with the majority of values falling between 0.173 and 0.186. The mean value for the prickly pear samples is 0.143 with the majority of values falling between 0.948 and 0.177. The majority of barrel cactus samples have values between 0.101 and 0.257, with a mean value of 0.134.

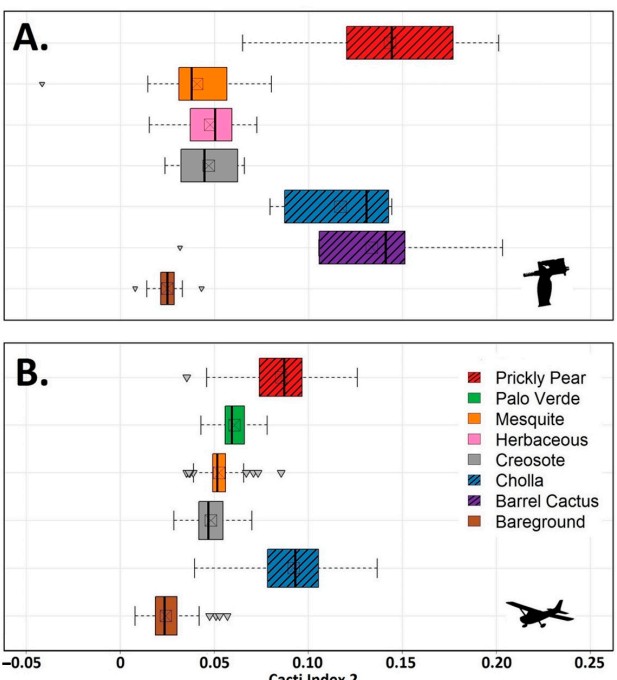

**Figure 4.** This graph shows the range of Cacti Index 2 (972 and 1072 nm) values for cacti, non-cacti vegetation, and bare ground computed from the Field Spectrometer (**A**) and NEON AVIRIS (**B**) collections. The red hatched boxes represent prickly pear, the blue hatched boxes represent cholla, and the purple hatched boxes represent barrel cactus. The square in the middle of each box is the mean for that series of samples. The whiskers represent the minimum and maximum values within the interquartile range. Any triangles outside the whiskers are outlier values.

Using the same examples of bare ground (N = 300), creosote (N = 418), herbaceous (N = 100), mesquite (N = 105), palo verde (N = 100), and prickly pear (N = 325) extracted from the NEON AVIRIS data we computed CI2. The calculated CI2 values for the selected pixels show a separation of the cholla and prickly pear (cacti) from non-cacti vegetation and bare ground. The majority of values for the cholla samples range from 0.078 to 0.106, while the majority of values for the prickly pear samples range from 0.074 to 0.097. The mean value for the cholla samples is 0.092 and the mean value for the prickly pear samples is 0.086 (Figure 4).

## 4. Discussion

The spectral signatures created from the direct measurements performed with the hand-held spectroradiometer demonstrate the unique features of cacti. It is clear that a water absorption dip occurs around 972 nm and reflectance peaks occur around 862 nm and 1072 nm for barrel, cholla, and prickly pear cacti. The dip is also present in the data of drone-mounted and airplane-mounted hyperspectral sensors. Nuanced differences between the aerial spectra and the hand-held spectra could be caused by a variety of factors, including 1. spatial resolution, 2. radiometric sensitivity, 3. The reflectance calculation method, and 4. atmospheric water absorption of spectra around 972 nm [31].

We observed diminished sensitivity of the Cacti Indices within the NEON AVIRIS data. This was probably a function of the coarser spatial resolution of that imagery. Within 1 m pixels, instead of being pure cacti reflectance, the features are often a mix of vegetation, bare ground, and shadow. Prickly pear samples were especially prone to mixed signals due to their spreading structural form (Figure 5). However, prickly pears tended to be large, 2–4 m in diameter, making up for shadowing created by their structural composition. Barrel cactus and cholla usually had less mixed signals due to their more compact morphology with diameters as small as 50 cm. Despite reduced sensitivity from AVIRIS NEON data, the cacti indices still demonstrated the ability to separate cacti from non-cacti vegetation. Though both CI1 and CI2 show separability between cacti and non-cacti vegetation, CI1 shows a broader separation, making it the preferred index in our study area.

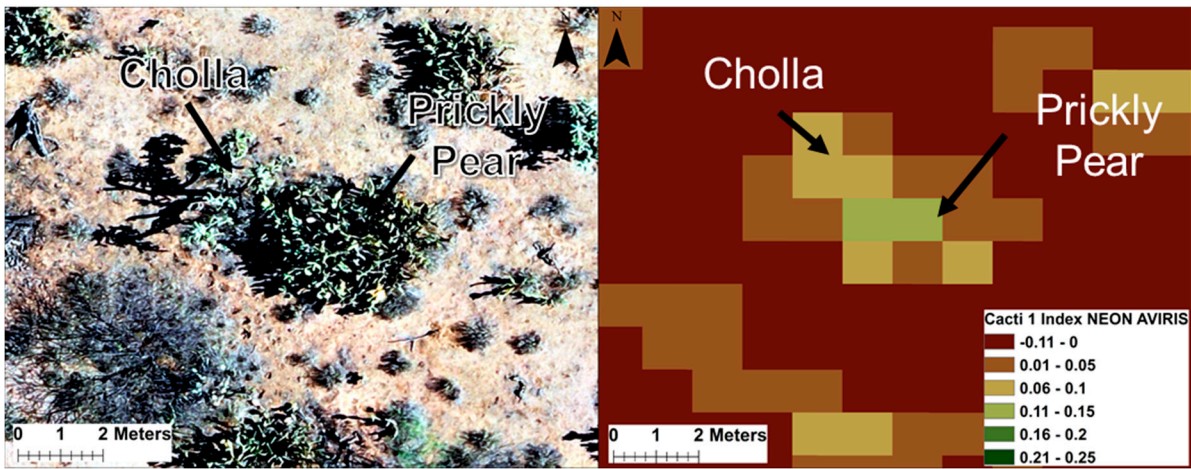

**Figure 5.** On the **left**: Example of a prickly pear cactus and a cholla cactus as captured by the DJI Mavic Pro Quadcopter. The nominal spatial resolution of the image is 1.6 cm. The difference in structure between the cholla and prickly pear is clear as seen by the influence of shadowing on the prickly pear cactus. On the **right**: The Cacti Index 1 visualized using the 1 m NEON AVIRIS reflectance data. It can be seen that cholla and prickly pear have higher cacti index values.

The Nano Hyperspectral sensor has diminished radiometric sensitivity near the edges of its range (near 400 nm and 1000 nm), which leads to a lower signal to noise ratio than other spectral bands in the sensor. This hardware limitation impacts CI1 produced from the Nano Hyperspectral sensor because 972 nm is near the edge of the silicon-based sensitivity.

The implication is that in addition to identifying cacti, the index produces many false high values for low light shadowed areas (Figure 6). Mitigation strategies could include identifying and removing low radiance pixels found in shadowed areas, or collecting imagery with longer exposure time. Ideally, drone-based mapping of cacti should use a hyperspectral sensor with a wider range (i.e., >1000 nm) than the Nano Hyperspectral sensor provides.

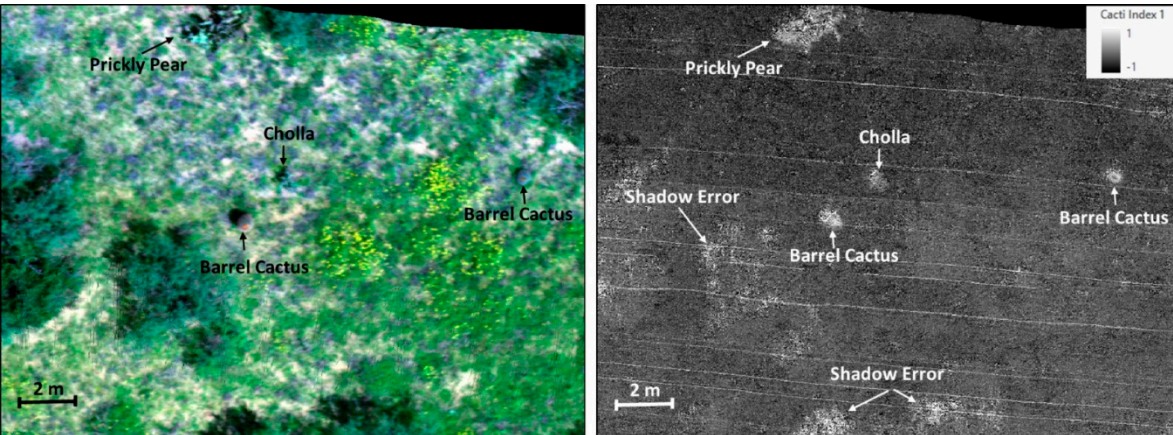

**Figure 6.** Drone-mounted Nano hyperspectral imagery shown as true color (**left** panel) and Cacti Index 1 (**right** panel). Cacti were easily identifiable with the index; however, sensitivity problems near the edge of the sensor range created false high index values in low light shadowed areas of the study area.

The spectral slices captured by the AVIRIS NEON sensor (5.5 nm) and Nano Hyperspectral sensor (2.4 nm) played a role in cacti detection. The smaller the slices the better the sensor is at capturing the difference between the peak at 862 nm and the dip at 972 nm. This led to a smaller dynamic range in the CI1 values calculated with the AVIRIS NEON data compared to the CI1 values calculated with the Nano Hyperspectral data.

The water absorption dip (972 nm) observed in the drone-based and airplane-based imagery demonstrate that the Cacti Indices can be used for the identification and mapping of cacti species across larger extents. Depending on the application, the Cacti Indices could be combined with other sensor data (e.g., LiDAR height information), used within a supervised classification framework, or implemented with a user-defined threshold.

The Cacti Indices should be effective in identifying succulent cacti in other regions of the world if the plant species store water in their tissue similar to barrel, cholla, and prickly pear cacti. However, localized research should investigate the extent to which non-target species might exhibit high index values and confuse cacti identification. In our study area for example, we discovered that Palo Verde, with their photosynthesizing stems, exhibited CI1 values nearly as high as prickly pear samples.

Using the Cacti Indices with satellite imagery may be possible but will be challenging. Our methods require narrow spectral bands and high spatial resolution to identify individual cacti. Additionally, the spectral area of interest near 972 nm is greatly impacted by water vapor in the atmosphere [31]. As a result, many satellite sensors do not have bands sensitive to this spectral region, and if they do, the signal will be quite weak. The Earth Observing-1 (EO-1) Hyperion (2000–2017) and a few other orbital hyperspectral reflectance sensors (HISUI-Hyperspectral Imager Suite-onboard the International Space Station; EnMAP-Environmental Mapping and Analysis Program; PRISMA-Hyperspectral Precursor and Application Mission) could possibly leverage the Cacti Indices [32]. These hyperspectral sensors all have moderate spatial resolutions (e.g., 30 m) that are unable to detect individual cacti, but could be used to estimate percent cover of cacti per pixel. More research on these sensors and platforms is needed.

## 5. Conclusions

In this paper, we demonstrated the ability to create an index from hyperspectral data to accurately identify cacti. We used a hand-held spectroradiometer and two imaging spectrometers to collect spectral information from cholla, barrel, and prickly pear cacti plants and other prevalent landscape features. Based on those collections, we identified three unique aspects in the spectral reflectance signatures for cacti at 862 nm, 972 nm, and 1072 nm bands of the electromagnetic spectrum. Using those three portions of the cacti spectral signatures, we calculated two normalized difference indices: Cacti 1 (862 nm and 972 nm) and Cacti 2 (972 nm and 1072 nm). We then used hyperspectral data captured by drone and airplane to show the applicability of the Cacti indices at various spatial resolutions. Cacti samples showed spectral uniqueness in both the 3 cm drone hyperspectral imagery and the 1 m aerial hyperspectral imagery using the cacti indices.

Whether for conservation or control applications, the cacti indices derived from aerial platforms can help identify cacti across larger landscapes than is possible with field-based measurements. Hyperspectral data provide more precise spectral observations of plant characteristics than multi-spectral imagery sources. Though hyperspectral imagery availability is currently limited, interest in the technology is strong across land management, agriculture, and mining industries. Availability of hyperspectral imagery at multiple airborne and spaceborne scales is likely to proliferate in the near future.

**Author Contributions:** Conceptualization, W.J.D.v.L. and C.C.; methodology, W.J.D.v.L., C.C. and K.H.; software, K.H., C.C. and J.K.G.; validation, W.J.D.v.L., C.L.N., C.C., K.H. and J.K.G.; formal analysis, W.J.D.v.L., C.L.N., C.C., K.H. and J.K.G.; investigation, W.J.D.v.L., C.L.N., C.C., K.H. and J.K.G.; resources, W.J.D.v.L.; data curation, C.L.N., K.H. and J.K.G.; writing—original draft preparation, K.H.; writing—review and editing, W.J.D.v.L., K.H. and J.K.G.; visualization, K.H. and J.K.G.; supervision, W.J.D.v.L.; project administration, W.J.D.v.L.; funding acquisition, W.J.D.v.L. All authors have read and agreed to the published version of the manuscript.

**Funding:** This research received no external funding. Technology and Research Initiative Fund (TRIF) funding was provided to purchase the hyperspectral drone.

**Data Availability Statement:** Data can be obtained by contacting the lead author.

**Acknowledgments:** The authors appreciate the research support provided by the Arizona Remote Sensing Center. We are thankful to Angela Chambers, who helped with data acquisition and image interpretation. We downloaded the NEON AVIRIS hyperspectral data from the NSF NEON Data Portal.

**Conflicts of Interest:** The authors declare no conflict of interest. The funders had no role in the design of the study; in the collection, analyses, or interpretation of data; in the writing of the manuscript, or in the decision to publish the results.

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
