# Peer review of "A Novel Spectral Index to Identify Cacti in the Sonoran Desert at Multiple Scales Using Multi-Sensor Hyperspectral Data Acquisitions"

_land, doi:10.3390/land11060786_

Round 1

Reviewer 1 Report

What time of day were the flights

Line 149 – Was there any ground validation? Or is it very easy to distinguish plant type based on the RBG images?

Figure 3- it would be nice to know the sample size for each group. Its also hard to distinguish the dots from the x without having to zoom in. If I printed the paper they would look the same. I’d also make the y-axis text larger as you cant read it without zooming in. The font size in fig 4 is much better

Fig 4. Should have sample size

Fig 5, text is too small in figure. I cant read the legend.

Line 243 – what is causing the shadowed areas, is this clouds?

Figure 6 – Whats the rest of the green stuff in the image, is it all vegetated?

I feel like the paper would be better if you actually did image classification using the Cacti Indices and then present a level or accuracy with that. Perhaps compare the accuracy with the different image types. It seems like a bunch of the plants overlap, and quickly looking at the whisker plots its hard to tell which plants are cacti and which are not. Maybe have one group shaded and the other group hatched.

Author Response

Responses to Reviewer 1

All responses in bold.

What time of day were the flights?

The NEON AVIRIS (aircraft) data was captured between 9am and 12pm Arizona time. The Nano Hyperspec (drone) data was captured between 11am and 1pm Arizona time to minimize shadows.

Line 149 – Was there any ground validation? Or is it very easy to distinguish plant type based on the RBG images?

It is very easy to distinguish plant type based on the RGB images as demonstrated in Figure 5.

Figure 3- it would be nice to know the sample size for each group. Its also hard to distinguish the dots from the x without having to zoom in. If I printed the paper they would look the same. I’d also make the y-axis text larger as you cant read it without zooming in. The font size in fig 4 is much better

Sample sizes have been added in the text at lines 176, 183, and 190 for the three types of sensors. Figure 3 has been transposed. The dots have been changed to triangles. The entire figure has been enlarged along with the text and legend. Cacti vegetation is now hatched.

Fig 4. Should have sample size

Sample sizes have been added in the text at lines 214 and 221 for the two types of sensors. Figure 4 has been transposed and cacti vegetation is now hatched.

Fig 5, text is too small in figure. I cant read the legend.

Increased font size of legend, scale bar, and labels.

Line 243 – what is causing the shadowed areas, is this clouds?

The shadows referenced in this line are related to the height of plants which leads to artifacts on the edges of taller plants. Clouds did not impact this study.

Figure 6 – Whats the rest of the green stuff in the image, is it all vegetated?

There are some perennial grasses present due to recent rain, including Lehman’s lovegrass (Eragrostis lehmanniana) along with several species of annual forbs, and velvet mesquite (Prosopis velutina). You can see that the non-cacti vegetation does not register on the Cacti Index image (right side of the figure).

I feel like the paper would be better if you actually did image classification using the Cacti Indices and then present a level or accuracy with that. Perhaps compare the accuracy with the different image types. It seems like a bunch of the plants overlap, and quickly looking at the whisker plots its hard to tell which plants are cacti and which are not. Maybe have one group shaded and the other group hatched.

The authors thought a great deal about whether or not to take the next logical step to perform an image classification with the cacti indices. We decided to focus on the spectral signatures at each scale of observation and omit the classification in this paper. Performing a classification would muddle the objective of developing a spectral cacti index and it would be difficult to identify the indices as the key variables in the success of classifications. In practical terms, the cacti indices could be used in a supervised classification framework. We discuss this starting on line 271. The cacti species have been hatched on the box and whisker graphs to distinguish them from non-cacti vegetation and bare ground.

Reviewer 2 Report

Dear Authors,

the paper presents an interesting study. However, before publication, I suggest to make a few corrections.

1) Firstly, I think that the structure of the manuscript looks like to a technical note rather than a typical article.

2) Fig.2 should be enlarged and more recognizable colors should be used to differentiate the spectral signatures of the analyzed vegetation.

3) The methodology lacks in a clear explanation about the choice of the two reflectances complementing the pillar of 972nm fixed by Leeuwen (2009). For these reflectances (862nm 1072??) authors speak generically of peaks. Please explain in detail the reason why of this choice. 

4) Authors compare the efficacy of the cacti index computed using hyperspectral sensors mounted on a drone (3 cm spatial resolution) and airplane (1 m spatial resolution). It would be appropriate to say a few words on the comparability of the two used sensors.

5) To facilitate the understanding of the overall paper, it would be useful to indicate the typical dimensions of the cacti.

6) Authors should also consider to spend a word about the interference of possible shadows within the analyzed scenes?

7) Major results could be usefully synthetized in a table reporting the comparison between drone and airplane.

Kind regards

Author Response

Responses to Reviewer 2

All responses in bold.

1) Firstly, I think that the structure of the manuscript looks like to a technical note rather than a typical article.

The authors will accept this designation if the editor deems it appropriate.

2) Fig.2 should be enlarged, and more recognizable colors should be used to differentiate the spectral signatures of the analyzed vegetation.

The authors enlarged Fig.2 to the size of a full page. The width of the lines has been increased.

3) The methodology lacks in a clear explanation about the choice of the two reflectances complementing the pillar of 972nm fixed by Leeuwen (2009). For these reflectances (862nm 1072??) authors speak generically of peaks. Please explain in detail the reason why of this choice. 

The authors adjusted this section starting at line 111 to read, “We measured the magnitude of the dip with a normalized difference approach between the reflectance values at the bottom of the dip and reflectance values immediately outside of the dip. This is similar to the concept of the normalized difference vegetation index (NDVI)[29]. A priori, it was unknown which specific bands outside of the dip would provide a robust index capable of distinguishing cacti from non-cacti vegetation. We tested two different spectral indices.”

4) Authors compare the efficacy of the cacti index computed using hyperspectral sensors mounted on a drone (3 cm spatial resolution) and airplane (1 m spatial resolution). It would be appropriate to say a few words on the comparability of the two used sensors.

The authors added the following paragraph starting at line 266, “The spectral slices captured by the AVIRIS NEON sensor (5.5nm) and Nano Hyperspec sensor (2.4nm) played a role in cacti detection. The smaller the slices the better the sensor is at capturing the difference between the peak at 862nm and the dip at 972nm. This led to a smaller dynamic range in the CI1 values calculated with the AVIRIS NEON data compared to the CI1 values calculated with the Nano Hyperspec data.”

5) To facilitate the understanding of the overall paper, it would be useful to indicate the typical dimensions of the cacti.

The authors added the following starting at line 240, “However, prickly pears tended to be large, 2 – 4 meters in diameter, making up for shadowing created by their structural composition. Barrel cactus and cholla usually had less mixed signals due to their more compact morphology with diameters as small as 50 centimeters.”

6) Authors should also consider spending a word about the interference of possible shadows within the analyzed scenes?

The authors address this on lines 237 – 241 and lines 256-259.

7) Major results could be usefully synthetized in a table reporting the comparison between drone and airplane.

The authors recognize the utility of a summary table, but do not feel the results of this research can be summarized in that manner.

Round 2

Reviewer 1 Report

Overall I think that the improvements the author made were sufficient, though I feel like the paper would be greatly enhanced if it also included image classification based on the spectral dip the authors found in their study.